# Doppler Indices of the Uterine, Umbilical and Fetal Middle Cerebral Artery in Diabetic versus Non-Diabetic Pregnancy: Systematic Review and Meta-Analysis

**DOI:** 10.3390/medicina59081502

**Published:** 2023-08-21

**Authors:** Sonja Perkovic-Kepeci, Andja Cirkovic, Natasa Milic, Stefan Dugalic, Dejana Stanisavljevic, Milos Milincic, Konstantin Kostic, Nikola Milic, Jovana Todorovic, Ksenija Markovic, Natasa Aleksic Grozdic, Miroslava Gojnic Dugalic

**Affiliations:** 1General Hospital Pancevo, 26000 Pancevo, Serbia; perkovicsonja@yahoo.com; 2Faculty of Medicine, University of Belgrade, 11000 Belgrade, Serbia; k.ginekos@gmail.com (K.K.); nmilic@yahoo.com (N.M.); xeniakm3@gmail.com (K.M.); miroslavagojnicdugalic@yahoo.com (M.G.D.); 3Institute for Medical Statistics and Informatics, Faculty of Medicine, University of Belgrade, 11000 Belgrade, Serbia; natasa.milic@med.bg.ac.rs (N.M.); dejana.stanisavljevic@med.bg.ac.rs (D.S.); 4Division of Nephrology and Hypertension, Mayo Clinic, Rochester, MN 55905, USA; 5Clinic for Gynecology and Obstetrics, University Clinical Centre of Serbia, 11000 Belgrade, Serbia; stef.dugalic@gmail.com (S.D.); milosmilincic@gmail.com (M.M.); 6Institute of Social Medicine, Faculty of Medicine, University of Belgrade, 11000 Belgrade, Serbia; jole6989@hotmail.com; 7Institute for Process Engineering Environmental Engineering and Technical Life Sciences, Technical University of Vienna, 1180 Vienna, Austria; natasaaleksicviii2@yahoo.com

**Keywords:** diabetes mellitus, pregnancy, Doppler ultrasound, indices

## Abstract

*Background and Objectives*: The aim of this study was to assess the differences in Doppler indices of the uterine (Ut), umbilical (UA), and middle cerebral artery (MCA) in diabetic versus non-diabetic pregnancies by conducting a comprehensive systematic review of the literature with a meta-analysis. *Materials and Methods*: PubMed, Web of Science, and SCOPUS were searched for studies that measured the pulsatility index (PI), resistance index (RI), and systolic/diastolic ratio index (S/D ratio) of the umbilical artery, middle cerebral artery, and uterine artery in diabetic versus non-diabetic pregnancies. Two reviewers independently evaluated the eligibility of studies, abstracted data, and performed quality assessments according to standardized protocols. The standardized mean difference (SMD) was used as a measure of effect size. Heterogeneity was assessed using the I2 statistic. Publication bias was evaluated by means of funnel plots. *Results*: A total of 62 publications were included in the qualitative and 43 in quantitative analysis. The UA-RI, UtA-PI, and UtA-S/D ratios were increased in diabetic compared with non-diabetic pregnancies. Subgroup analysis showed that levels of UtA-PI were significantly higher during the third, but not during the first trimester of pregnancy in diabetic versus non-diabetic pregnancies. No differences were found for the UA-PI, UA-S/D ratio, MCA-PI, MCA-RI, MCA-S/D ratio, or UtA-RI between diabetic and non-diabetic pregnancies. *Conclusions*: This meta-analysis revealed the presence of hemodynamic changes in uterine and umbilical arteries, but not in the middle cerebral artery in pregnancies complicated by diabetes.

## 1. Introduction

Pre-gestational (DM) and gestational diabetes mellitus (GDM) are associated with increased risk of adverse perinatal outcomes [1,2]. Maternal hyperglycemia provokes in utero adaptation by fetal hyperinsulinemia, which causes increased nutrient storage, and in turn the development of fetal macrosomia. Fetal macrosomia complicates delivery and might put mother and baby at risk of birth injuries [3]. In a large meta-analysis including 7.5 million pregnancies, GDM was significantly associated with a range of adverse pregnancy outcomes [4]. Women with GDM and no insulin use have increased odds of caesarean section, preterm delivery, macrosomia, infant born large for gestational age and low Apgar score, while for women with GDM using insulin, the odds of having an infant requiring admission to the neonatal intensive care unit, who is large for gestational age, with respiratory distress syndrome, and/or neonatal jaundice were higher than in those without GDM [4]. It is currently common practice to consider earlier labor inductions based on glycemia status in order to reduce this risk of adverse outcomes in pregnancies complicated by GDM [5,6]. It was also shown that a correct pregnancy diet and maternal weight gain could modify the hyperglycemia status and reduce the risk of GDM and its complications [7] and that even moderate changes in pre-pregnancy weight can apparently affect the risk of GDM among obese women [8].

Since hyperglycemia starts its effect during organogenesis, this condition is also known as diabetic embryopathy [9]. It affects the cardiovascular, central nervous, gastrointestinal, genitourinary, and musculoskeletal system, and 6–12% of fetuses with diabetic embryopathy would have congenital problems of this kind [10]. Diabetic embryopathy is also known to be associated with a higher rate of miscarriages [11]. Hyperglycemia creates anaerobic in utero setting, leading to hypoxia and acidosis, which could result in a stillbirth [12]. Complications reported from newborns delivered from diabetic pregnancies are neonatal hypoglycemia, hyperbilirubinemia, hypocalcemia, polycythemia, respiratory distress syndrome [13], as well as increased risk for obesity, diabetes, and hypertension in developing years [14].

During a physiological pregnancy, spiral remodeling modifies arteries from low-flow/high-resistance to high-flow/low-resistance vessels [15,16], but the maternal diabetes may change this process and the functioning of the placenta. Poor nutrient and oxygen transfer across the placenta lead to fetal hypoxia [17], while delayed metabolic products removal increases the risk of fetal asphyxia [18]. Hyperglycemia, both in fetus and mother, leads to changes in vascular condition, higher oxidative stress, and awakening of epigenetic remodeling [19,20]. Changes on the placental level are angiomorphological and pathophysiological with implications on hemodynamics, reducing utero-placental perfusion. The protection mechanism for the fetus is known as the “brain sparing” phenomenon. Blood from the peripheral blood stream is being redistributed to the brain instead of the viscera, which can be seen in a decreased fetal middle cerebral artery resistance and pulsatility index and increased umbilical artery resistance and pulsatility index [21,22]. These hemodynamic changes could be revealed by Doppler ultrasound measurements [18,23]. The predictive power of Doppler US for adverse perinatal outcomes in both high- and low-risk pregnancies has been proven by numerous studies [16]. It still remains uncertain to which extent altered hemodynamics accompanies diabetic pregnancies. Therefore, the aim of this study was to assess the differences in pulsatility (PI), resistance (RI) and systolic/diastolic ratio (S/D ratio) Doppler indices of uterine (Ut), umbilical (U), and middle cerebral artery (MCA) in diabetic versus non-diabetic pregnancies by conducting a comprehensive systematic review of the literature with a meta-analysis.

## 2. Materials and Methods

### 2.1. Study Design

This systematic review was registered at PROSPERO (CRD42023409966) and is conducted according to the PRISMA protocol recommendations (Reporting Items for Systematic Reviews and Meta-Analyses) [24] and MOOSE guidelines [25].

### 2.2. Eligibility Criteria

Original studies that measured Doppler indices (pulsatility, resistance, and systolic/diastolic ratio) of umbilical, uterine, and middle cerebral arteries in pregnant women with pre-gestational or gestational DM were included. The inclusion criteria were developed according to the PICOS system: (P) population: all pregnant women; (E) exposure: pre-gestational DM or GDM; (C) control: non-DM or non-GDM; (O) outcome: pulsatility index (PI), resistance index (RI), or systolic/diastolic ratio (S/D ratio) of umbilical, uterine, and middle cerebral arteries; (S) study design: controlled trials, prospective or retrospective cohort, nested case-control in cohort studies, case-control studies, and cross-sectional studies.

The exclusion criteria were: (i) language: other than English; (ii) not an original article: narrative reviews, systematic reviews, meta-analysis, case reports, case series, editorials, comments, correspondences, books, short, abstracts, etc.; (iii) wrong population: other than humans (animals, cell lines), not pregnant women; (iv) no control group; (v) inadequate control group: not non-DM pregnant women; (vi) wrong outcome: other indices than PI, RI, and S/D ratio for arteries other than umbilical, uterine, and cerebral medial artery.

Two researchers with expertise in conducting systematic reviews and meta-analyses (AC, NM) developed and ran the search. The following databases were electronically searched: PubMed, Web of Science (WoS), and SCOPUS until 6 September 2022. The following search queries were combined to identify all relevant articles that measured Doppler indices among pregnant women with GDM and pregnant women with pregestational-GDM: (Gestational diabetes mellitus and (Color Doppler ultrasonography or Color Doppler ultrasonography or Doppler or Doppler sonography or Doppler velocimetry or Pulse wave Doppler or pulsatility index or peak systolic velocity or systolic/diastolic ratio or S/D ratio or resistance index or resistive index or resistivity index)) or (Diabetes mellitus and pregnancy and (Color Doppler ultrasonography or Doppler or Doppler sonography or Doppler velocimetry or Pulse wave Doppler or pulsatility index or peak systolic velocity or systolic/diastolic ratio or S/D ratio or resistance index or resistive index or resistivity index)) (details are available in Appendix A). In addition, reference lists of articles identified through electronic search and relevant reviews and editorials were manually searched to check for more potentially relevant articles.

### 2.3. Article Screening and Selection

Publications were screened for inclusion by title and abstract reading independently by two reviewers (M.M., K.K.) in the first step, and by full-text reading by two new reviewers (S.P.-K., A.C.). All disagreements were resolved by discussion at each stage with the inclusion of a third reviewer if needed (M.G.D. or D.S. or N.M.). A Rayyan online application was used for the first step of the selection process. Studies were included in the full-text screening if the study was identified as potentially eligible or if the abstract and title did not have sufficient information for exclusion.

### 2.4. Data Abstraction and Quality Assessment

Two reviewers (S.P., A.C.) independently abstracted the following data: (i) authors, publication year, country, study design, measured Doppler index, and artery; (ii) type of DM, sample size, characteristics of cases and controls, glycaemia, HbA1c, maternal age, gestational age, body weight, body mass index; (iii) criteria for DM; (iv) inclusion and exclusion criteria for cases and controls; and (v) newborns gender, body weight, Apgar score in the 1st and 5th minute. Previously designed protocol was used for data extraction. Authors of relevant articles were contacted to obtain unavailable manuscripts and/or missing data. Each reviewer independently performed a risk of bias and quality assessment of the included articles using an adapted version of the Newcastle-Ottawa tool (NOS) for observational studies [26]. The study quality, according to NOS, was defined: good (3 or 4 stars in selection AND 1 or 2 stars in comparability AND 2 or 3 stars in outcome/exposure domain, or ≥7 stars in total), fair (2 stars in selection AND 1 or 2 stars in comparability AND 2 or 3 stars in outcome/exposure domain, or 5–6 stars in total), or poor (0 or 1 star in selection OR 0 stars in comparability OR 0 or 1 star in outcome/exposure, or ≤4 stars in total). Results of the quality assessment is given in Appendix A.

### 2.5. Statistical Analysis

The primary outcome was the difference in the PI, RI, and S/D ratio Doppler indices of the umbilical, uterine, and middle cerebral artery in diabetic versus non-diabetic pregnancies. While figures were used to present Doppler indices, GraphGrabber was used to read indices values. If data were not presented as an arithmetic mean with standard deviation, the following approximations were used: (1) if median was available, median was used as an approximation of the mean; (2) where z score was available, the mean was calculated according to the following formula [27]: (sd × z) where sd = se × √n; (3) if the multiple of median (MoM) was available, mean was calculated as MoM = median(patient/population value) [27]; (4) if IQR was available, standard deviation (sd) was calculated as sd = IQR/1.35; (5) if standard error (se) was used, sd was obtained by the following formula sd = se × √n; (6) if range was reported, sd was calculated as sd = (max − min)/4, and; (7) if 95%CI was used, sd was calculated as ((Upper limit of 95%CI − ((Upper limit of 95%CI + Lower limit of 95%CI)/2))/1, 96) × √n.

The standardized mean difference (SMD) was used to examine differences in diabetic versus non-diabetic pregnancies, due to different methodologies used for Doppler measurements across the studies included in the meta-analysis. SMD expresses the difference between group means in units of standard deviation and was estimated by pooling individual trial results using random-effects models via the Der Simonian-Laird method. Heterogeneity was assessed using the Chi-square Q and I2 statistic. I2 presents the inconsistency between the study results and quantifies the proportion of observed dispersion that is real, i.e., due to between-study differences and not due to random error. The categorization of heterogeneity was based on the Cochrane Handbook [28]: I2 < 30%, 30–60%, or >60%, correspond to low, moderate, and high heterogeneity, respectively. Forest plots were constructed for each analysis showing the SMD (box), 95% confidence interval (lines), and weight (size of box) for each study. The overall effect size was represented by a diamond. Publication bias was assessed by funnel plots for each defined outcome (Appendix A). Subgroup analysis was performed for (1) pregestational and gestational DM and (2) Doppler indices measured in the 1st, 2nd, and 3rd trimester separately. Sensitivity analyses were conducted to examine the effects of: (1) different DM cases (removing the combination of DM and other diseases like PE, HPD). *p* value ≤ 0.05 was considered statistically significant. Analyses were performed using Review Manager Version 5.4.

## 3. Results

### 3.1. Systematic Review

A total of 10,820 potentially eligible articles were found. After removal of 6983 duplicates, 3837 articles were screened for inclusion based on the title and abstract reading. After the exclusion of 3686 articles (due to wrong publication type, population, outcome, method, no presence of control group or language other than English), 151 publications were screened for inclusion based on full-text reading. A total of 62 articles were selected for inclusion in the qualitative and 43 for quantitative synthesis. A flow chart illustrating the selection process is presented in Figure 1.

Characteristics of all publications included in the systematic review are presented in detail in Table 1. Studies were published between 1987 and 2022, with a total of 156,166 participants; 9912 women with and 146,254 without DM. The minimum and maximum sample size of the DM group was 9 and 4015, while for the non-DM group it was 10 and 71,565. Matching was applied in 23% of studies only; gestational age at the time of delivery and maternal age were the most commonly used variables for matching (in 9/15 and 6/15 studies, respectively). Other matching variables were: obesity, weight gain during pregnancy, BMI at the time of delivery, chronic hypertension, parity, race, gravidity, past obstetric history, and smoking. Prospective cohort studies were the most common among included studies (20/62); 8 studies were cross-sectional, 7 studies were case-controls, and 1 study was a retrospective cohort. Eleven studies did not correctly report study design and 15 did not report study design at all. Most studies were performed in Europe (23) and Asia (22). There were also studies from North America (9), Africa (4), South America (2), and Australia and Oceania (2). The predominant population included in studies were pregnant women with GDM (39/62). Pregnant women with pre-GDM type 1 were assessed in 21/62 studies, pre-gestational diabetes mellitus type 2 in 9/62, while the type of pre-GDM was not specified in 9 studies. Doppler ultrasonography was performed during the 3rd trimester in 39/62 studies, 2nd trimester in 18/62, and 1st trimester in one study. The exact timing of Doppler measurements was not reported in 12 studies. The most assessed Doppler index was the pulsatility index (33/62); the resistance index was measured in 15 studies, while the S/D ratio index was used in 11 studies. All 33 studies that assessed PI performed measurement on the umbilical artery; PI was measured on the middle cerebral artery in 20/33 and on the uterine artery in 13/33. The umbilical artery RI was measured in 16 studies, middle cerebral artery RI in 10/15, and uterine artery RI in 3/15. The systolic/diastolic ratio index was measured in all 11 studies on the umbilical artery, while it was measured on the middle cerebral artery in 7/11, and on the uterine artery in 3/11.

A total of 41/62 of included studies reported specific criteria and 37/41 a definition for DM diagnosis as well. White’s classification of Diabetes in Pregnancy, World Health Organization (WHO), and American Diabetes Association (ADA) criteria were used in 11, 8, and 7 studies, respectively. Other criteria that were applied were: IADPSG (4), O’Sullivan (3), National Diabetes Data group (2), Australian Diabetes in Pregnancy (ADIPS) (2), National Institute for Health and Clinical Excellence (NICE) guidelines, American College of Obstetricians and Gynecologists (ACOG), Fifth International Workshop-Conference on Gestational Diabetes, and the Sixth edition of Obstetrics and Gynecology in one study each. Details regarding DM definitions and the diagnostic criteria used in the included articles are presented in Appendix A. The most common exclusion criterium was multiple pregnancy (31/62), while other exclusion criteria were: chronic hypertension (17/62), preeclampsia (14/62), pregnancy-induced hypertension (12/62), smoking (11/62), renal diseases (11/62), cardiovascular diseases (10/62), obesity (4/62), and nulliparity (1/62). Inclusion and exclusion criteria used in included studies are presented in detail in the Appendix A. The characteristics of newborns were rarely reported. Birth weight was available in 42/62 studies, gender in 14, while Apgar score was available in 18 studies. Appendix A presents newborns’ characteristics in more detail.

### 3.2. Meta-Analysis

A meta-analysis was performed for the UA-PI, UA-RI, UA-S/D ratio, MCA-PI, MCA-RI, MCA-S/D ratio, UtA-PI, UtA-RI, and UtA-S/D ratio Doppler indices. The UA-RI, UtA-PI, and UtA-S/D ratio were significantly higher in diabetic in contrast to non-diabetic pregnancies (SMD = 0.40, 95%CI = 0.07–0.73, *p* = 0.020 (Figure 2); SMD = 1.62, 95%CI = 0.36–2.88, *p* = 0.010 (Figure 3), and SMD = 1.02, 95%CI = 0.02–2.03, *p* = 0.050 (Figure 4), respectively).

Subgroup analysis showed increased levels of UtA-PI measured during the 3rd trimester (SMD = 0.47, 95%CI = 0.09–0.86, *p* = 0.020), but not during the 1st trimester of pregnancy (SMD = 0.65, 95%CI = −0.79–2.09, *p* = 0.380), in diabetic versus non-diabetic pregnancies (Figure 5).

The following Doppler indices were not significantly different in diabetic versus non-diabetic pregnancies: UA-PI (SMD = 0.12, 95%CI = −0.05–0.29, *p* = 0.170) (Appendix A), UA-S/D ratio (SMD = 0.01, 95%CI = −0.37–0.39, *p* = 0.960) (Appendix A), MCA-PI (SMD = 0.15, 95%CI = −0.12–0.42, *p* = 0.280) (Appendix A), MCA-RI (SMD = 0.21, 95%CI = −0.57–0.98, *p* = 0.600) (Appendix A), MCA-S/D ratio (SMD = −0.28, 95%CI = −1.07–0.51, *p* = 0.480) (Appendix A), and UtA-RI (SMD = 0.66, 95%CI = −0.40–1.73, *p* = 0.220) (Appendix A).

Sensitivity analysis including studies of gestational versus non-GDM pregnancies presented no significant differences in the following Doppler indices: UA-PI (SMD = 0.04, 95%CI = −0.10–0.19, *p* = 0.540) (Appendix A), UA-RI (SMD = 0.16, 95%CI = −0.08–0.41, *p* = 0.190) (Appendix A), UA-S/D ratio (SMD = 0.18, 95%CI = −0.19–0.54, *p* = 0.340) (Appendix A), MCA-PI (SMD = 0.15, 95%CI = −0.13–0.43, *p* = 0.300) (Appendix A), MCA-RI (SMD = 0.28, 95%CI = −0.71–1.27, *p* = 0.580) (Appendix A), MCA-S/D ratio (SMD = −0.28, 95%CI = −1.07–0.51, *p* = 0.480) (Appendix A), and UtA-PI (SMD = 0.63, 95%CI = −0.13–1.38, *p* = 0.100) (Appendix A).

## 4. Discussion

This is the first systematic review with a meta-analysis assessing differences in pulsatility, resistance, and systolic/diastolic ratio Doppler indices of the uterine, umbilical, and middle cerebral artery between pregnant women with and without diabetes mellitus. The UA-RI, UtA-PI, and UtA-S/D ratio had higher values in pregnant women with than without DM. Subgroup analysis showed that levels of UtA-PI were significantly higher in DM than in non-DM pregnant women during the 3rd, but not during the 1st trimester.

The maternal body goes through many physiological adaptations to fulfill pregnancy requirements. Healthy pregnancy is a state of mild insulin resistance that becomes obvious in the late 2nd trimester due to the dysfunction of beta cells in the mother’s pancreas resulting in higher blood glucose levels. These changes occur due to hormonal secretion of the placenta, weight gain, and endothelial dysfunction through enhanced inflammation and a Th-2 predominant immune response [87]. Intensive production of human placental lactogen, estrogen, progesterone, prolactin, and cortisol [19,88], as well as adipocytokines (leptin, tumor necrosis factor alpha, interleukin-6, resistin, visfatin, apelin, and retinol-binding protein 4) are contributing the most to disrupted glucose homeostasis during pregnancy [19,87]. Morphology changes in placenta in terms of infarctions, retroplacental hemorrhage, distal villous hypoplasia, and decidual arteriopathy are induced by the aforementioned processes [89]. The endothelial dysfunction together with higher blood glucose concentrations produce higher blood flow viscosity, thus the blood flow resistance increases while blood flow speed decreases, which easily leads to abnormal blood perfusion [18]. During the course of pregnancy, changes in the uteroplacental, fetoplacental, and fetal circulation, representing the oxygen metabolism in between the three compartments maternal, feto-maternal, and fetal, become more detectable [21]. Reference ranged Doppler values measured on uterine, umbilical, and cerebral media arteries are the mirror of efficient circulation necessary for adequate fetal development and growth [90].

Our study demonstrated increased UA-RI, UtA-PI, and UtA-S/D ratio Doppler indices in pregnant women with DM in contrast to those without DM. Previous studies reported inconsistent results regarding the arteries and Doppler indices measured, time of Doppler measurements, different forms of DM, and diabetes severity. Nicolaides et al. found no relation between UtA and UA with neither short-term nor long-term maternal glycemic control [91], and therefore concluded that impedance to flow in the uterine artery is normal in diabetic pregnancy, even in patients complicated with nephropathy and vasculopathy [91]. This was not the case with the umbilical artery, in the study by Gazzolo, where the increase in impedance was noticed in the state of maternal vasculopathy [39]. Abnormal UA-RI was associated with birthweights of less than 50th centile seen in diabetic pregnancy [33]. The same authors reported in 1992 that UA-RI declined significantly during the course of T1DM pregnancy [34] and in 1994 that UtA-RI was slightly higher in the presence of evident morphological vasculopathy [92]. Pietryga et al. [93] demonstrated significantly increased uterine artery vascular impedance in pregnant women with T1DM in cases with severe vasculopathy, while Gutaj et al. [94] obtained that the UA-RI increase does not depend on the level of vascular changes in the mother. UA-PI was the highest in pregnant women with T1DM in comparison with T2DM and GDM, while there was no difference in the mean MCA PI between these three groups [95]. Wei et al. [18] had found that the increase in the PI, RI, and S/D value during pregnancy were positively correlated with the onset of GDM, indicating that the arterial blood flow condition during pregnancy can reflect the formation process of GDM, and has certain clinical significance for GDM diagnosis and disease monitoring.

However, materno-fetal Doppler parameters can be affected not only by DM but by many other factors. Systemic diseases like hypertensive disorders in pregnancy and cardiovascular diseases have a lot of overlapping risk factors (age, smoking, obesity, etc.) with DM [96]. Inadequate vascular dilatation and angiogenesis are common pathohistological causes of hypertension in pregnancy, preeclampsia and GDM, denoting a failed response to the vasodilatory and pro-angiogenic challenge imposed by pregnancy, especially if multifetal [97]. GDM is also known to be a risk factor for later onset of gestational hypertension. The relationship between inadequate glucose milieu and higher blood pressure lies in reshaped uteroplacental vascularization [98], which results further on with abnormal uteroplacental blood flow [99]. In these cases, Doppler velocimetry measurements may have an important role in real-time antepartum surveillance as they have the ability to detect high-risk pregnancies in disrupted oxygenation states such as in hypoxemia, anemia, preeclampsia, IUGR, and DM [39,100,101]. It is also known that doppler velocimetry as a tool is very helpful in predicting adverse outcomes in twin pregnancies [102]. Although some of these factors like chronic diseases, preeclampsia, fetal growth retardation, and drug use, that may affect Doppler parameters, are stated as exclusion criteria in some studies included in our meta-analysis, the absence of such exclusion criteria (or not reporting them) in others may affect the results of our meta-analysis. Fouda et al. found that HgA1c was higher in pregestational diabetic women with chronic hypertension. Also, UA-RI was higher in diabetic pregnancies with hypertension, but not in diabetic pregnancies without hypertension, in comparison to uncomplicated pregnancies as controls [53]. Hssan et al. reported higher UA-PI levels in diabetic pregnancies complicated by preeclampsia [82]. In a recent study, tobacco combustion was associated with higher uterine and umbilical PI, RI, and S/D ratio Doppler indices with a strong association between indices values and the number of cigarettes smoked per day [103].

Results of our meta-analysis presented no significant differences between the DM and non-DM groups in terms of fetal MCA Doppler parameters. It is known that long-term uncontrolled hyperglycemia, chronic hypertension, preeclampsia, and IUGR can lead to placental vascular dysfunction with changes even in fetal circulation [53,82,104,105,106]. But the effect of metabolic changes due to diabetes mellitus during pregnancy on the fetus may be acidemia without hypoxemia, thus that redistribution seen in fetal hypoxemia may not occur even in severely compromised fetuses; and, therefore, it is of huge importance not to misrepresent this state by apparently normal fetal Doppler results [80].

This study has several limitations that should be considered when interpreting the results. First, the absence of exclusion criteria such as additional chronic diseases, preeclampsia and/or fetal growth retardation, and drug use in some of the included studies may affect the overall pooled estimate of this meta-analysis. Second, some patients with DM included in studies are followed by using insulin, and some are followed only by appropriate diet. This broad range of therapy regimens might also affect the results of the meta-analysis. Third, although the pregnancy trimesters are specified in some studies, it is possible that the differences between the gestational weeks of Doppler measurements applied in the studies affects the overall results.

## 5. Conclusions

This meta-analysis revealed the presence of hemodynamic changes in uterine and umbilical arteries, but not in middle cerebral artery in pregnancies complicated by diabetes. UtA-PI, UtA-S/D ratio, and UA-RI Doppler indices are higher in diabetic versus non-diabetic pregnancies. More studies are needed to distinguish effects of pregestational versus gestational diabetes on hemodynamic changes during pregnancy.

## Figures and Tables

**Figure 1 medicina-59-01502-f001:**
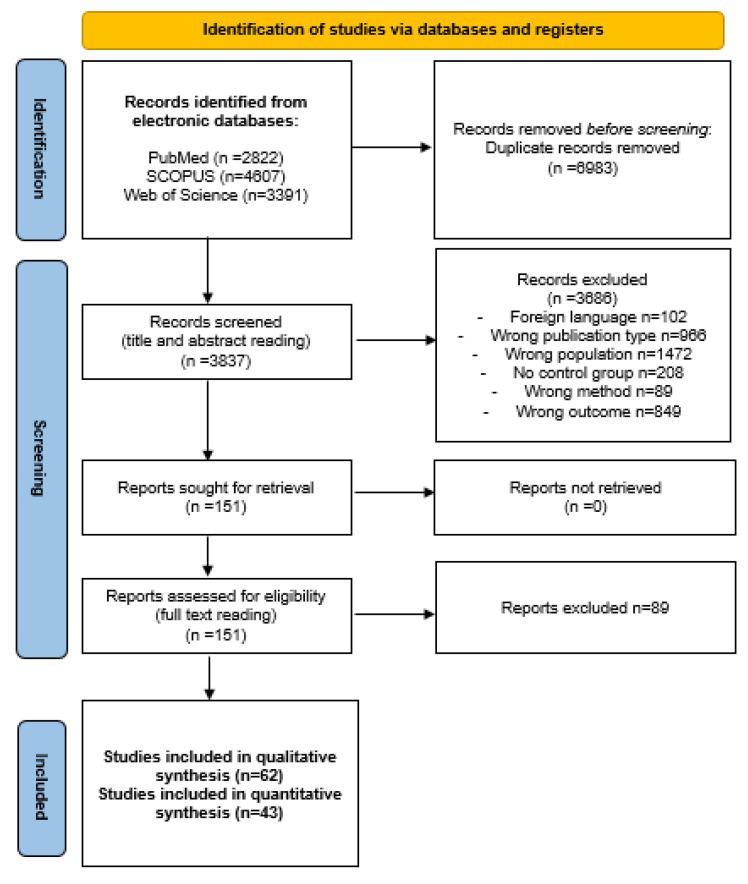
Flow diagram.

**Figure 2 medicina-59-01502-f002:**
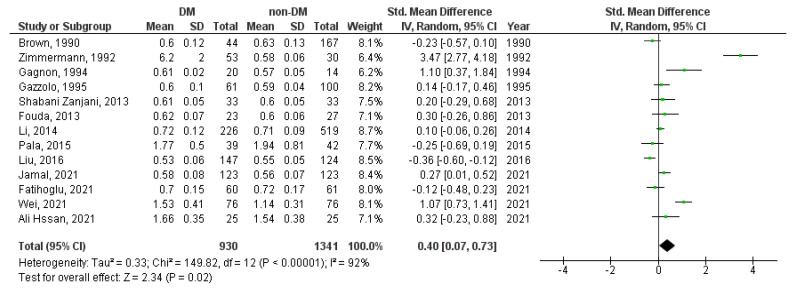
UA-RI Doppler index in diabetic versus non-diabetic pregnancies. The green squares represent each study individual SMD and the extending lines the 95% confidence intervals. The black diamond represents the overall estimate result [18,21,32,34,36,39,53,56,60,61,80,82,83].

**Figure 3 medicina-59-01502-f003:**
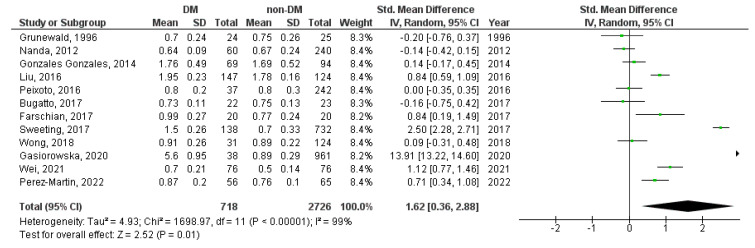
UtA-PI Doppler index in diabetic versus non-diabetic pregnancies. The green squares represent each study individual SMD and the extending lines the 95% confidence intervals. The black diamond represents the overall estimate result [14,18,41,52,57,61,62,63,64,67,70,84].

**Figure 4 medicina-59-01502-f004:**
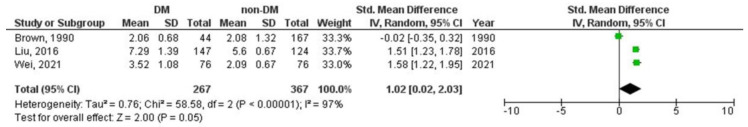
UtA-S/D ratio Doppler index in diabetic versus non-diabetic pregnancies. The green squares represent each study individual SMD and the extending lines the 95% confidence intervals. The black diamond represents the overall estimate result [18,32,61].

**Figure 5 medicina-59-01502-f005:**
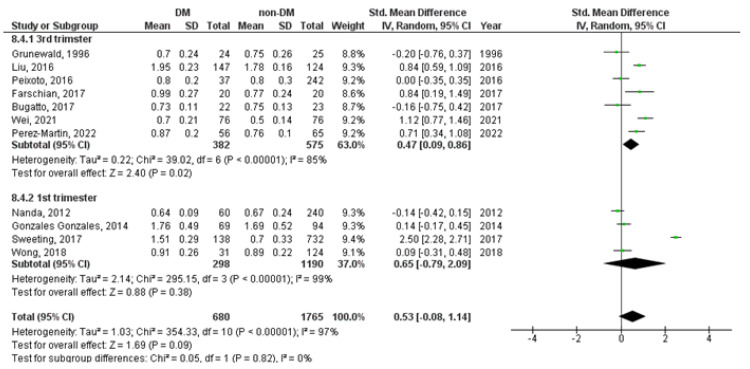
UtA-PI Doppler index in diabetic versus non-diabetic pregnancies according to the time of Doppler measurements. The green squares represent each study individual SMD and the extending lines the 95% confidence intervals. The black diamond represents the overall estimate result [14,18,41,52,57,61,62,63,64,67,70,84].

**Table 1 medicina-59-01502-t001:** Systematic review.

Study Characetristics			Cases						Controls					
Author (Year) Country Study Design	Artery-Index	DM Type (White Classification)	*n*	Characteristics	Glycaemia HbA1c	Maternal Age ^a^	Gestational Age ^b^	Weight BMI	*n*	Characteristics	Matched (Variable)	Maternal Age	Gestational Age	Weight BMI
Olofsson (1987) [29] Sweden NR	UA-PI	DM	40	Diabetic pregnancies	NR NR	29.2 (19–39)	26–34 (I test) 35–37 (II test) 37-delivery (III test) 37–42–term delivery in 37 <37–preterm delivery in 2 >42–post-term delivery in	NR NR	21	Healthy women with uncomplicated pregnancies	No	NR	NR	NR NR
Landon (1989) [30] USA NR	UA-S/D ratio	DM (B, C, D, F/R)	35	Insulin-dependent diabetic pregnant women	NR NR	NR	18–28 (at assessment)	NR NR	117	Normal non-diabetic pregnant women	No	NR	18–38 (at assessment)	NR NR
Friedman (1989) [31] USA prospective study	UA-S/D ratio	DM	18	Pregnant diabetic women with a genetic risk of heart disease or exposure to potential teratogens	NR NR	NR	16–38 (at assessment)	NR NR	113	Normal pregnant women	No	NR	14–41 (at assessment)	NR NR
Brown (1990) [32] Australia NR	UA-RI UA-S/D ratio UtA-RI UtA-S/D ratio	GDM	44	Diabetic pregnant women	NR NR	NR	>26 (at assessment) 38 ± 2 (at delivery)	NR NR	167	Normal pregnancies	No	NR	>26 (at assessment) 40 ± 2 (at delivery)	NR NR
Johnstone (1992) [33] UK prospective study	UA-PI	DM type 1 (B, C, D, F/R) GDM (A2)	128	Insulin-dependent diabetic pregnant women	NR NR	NR	>28 (at assessment)	NR NR	119	Non-diabetic pregnant women	No	NR	>28 (at assessment)	NR NR
Zimmermann (1992) [34] Finland prospective study	UA-RI	DM type 1 (B, C, D, F/R)	53	Insulin-dependent diabetic pregnant women	6.2 ± 2.0 mmol/L 6.6 ± 1.1%	26.8 ± 5.6	>17 (at assessment) 37.7 ± 1.3 (at delivery)	NR >27 kg/m^2^ in 11 (21%) women	30	Non-diabetic normal pregnancies at 37–38 weeks gestation with subsequently normal fetal outcome	No	NR	37–38 (at assessment)	NR NR
Pachi (1993) [35] Italy NR	UA-PI	DM type 1 (B, C, D, R)	30 Total Group I–10 Group II–10 Group III–10	Insulin-dependent diabetic pregnant women	Group I (<6.7 mmol/L): 5.5 ± 0.5 mmol/L Group II (6.1–7.2 mmol/L): 6.8 ± 0.3 mmol/L group III (>7.2 mmol/L) 8.1 ± 0.6, mmol/L NR	Group: I 30.3 ± 3.0 Group: II 29.0 ± 3.1 Group: III 29.2 ± 3.8	31 and 34 (test 1 and 2)	Pre-pregnancy weight (kg) Group I 54.3 ± 3.0. Group II 55.2 ± 3.3 Group III 56.7 ± 3.4 NR	150	Healthy pregnant women	No	NR	NR	NR NR
Gagnon (1994) [36] Canada NR	UA-RI	DM type 1 DM type 2 GDM	37 Total 17 DM (16 type I + 1 type II) 20 GDM	Diabetic pregnant women without diabetic retinopathy or nephropathy	DM (mean ± se) 128.0 ± 1.4 mg/dL, at 30 gw, 108.2 ± 1.3 mg/dL at 38 gw GDM (mean ± se) 121.5 ± 5.4 mg/dL at 30 gw, 109.0 ± 2.9 mg/dL, at 31 gw NR	NR	30, 33, 36 (at 1st assessment and weekly thereafter until delivery) DM mean (range) 38.2 (35–40) GDM 38.5 (36–40)	NR >27.3 kg/m^2^ in 3 DM and 18 GDM	14	Pregnant women with normal glucose metabolism defined as both screening tests negative (at 28 and 36 gw)	No	NR	40.1 (37–41) mean (range) (at delivery)	NR >27.3 kg/m^2^ in 1 control
Weber (1994) [37] USA NR	UA-S/D ratio	DM type 1 (B, C, D, F, RF)	9	Well-controlled insulin-dependent diabetic pregnant women without HTA or PE	NR 4.5 ± 0.6%, (20–26 gw) 4.6 ± 0.9%, (27–33 gw) 4.1 ± 0.3%, (34–40 gw) 3.8 ± 0.3%, (at delivery)	NR	20–26 (test 1) 27–33 (test 2) 34–40 (test 3) 38.1 ± 1.06 (at delivery)	NR NR	11	Nondiabetic volunteers randomly selected with normal medical histories and normal oral glucose tolerance tests excluding gestational diabetes	No	NR	20–26 (test 1) 27–33 (test 2) 34–40 (test 3) 40.6 + 91.3 (at delivery)	NR NR
Santolaya (1994) [38] USA NR	UA-RI	GDM	10	Obese GDM women with preconceptional weight > 90.7 kg	NR NR	28.3 ± 1.8	>20 (at assessment) 37.6 ± 0.9 (at delivery)	over 70.9 kg NR	18 Total 9-1st control 9-2nd control	1st control—obese woman 2nd control—obese women with PIH	No	1st control-24.7 ± 1.7 2nd control-30.0 ± 3.4	>20 (at assessment 1st control-38.8 ± 0.7 (at delivery) 2nd control-36.9 ± 1.8 (at delivery)	over 70.9 kg NR
Gazzolo (1995) [39] Italy NR	UA-RI	GDM	71	GDM pregnancies: treated with diet and insulin–Group A and group with abnormal neonatal neurological outcome-Group B	Group A GLY I 6.35 ± 2.72 mmol/L (27–32 gw) GLY II 5.97 ± 2.60 mmol/L (33–36 gw) Group B GLY I 6.08 ± 1.41 (27–32 gw) GLY II 5.91 ± 1.72 mmol/L (33–36 gw) NR	NR	27–32 (test 1) 33–36 (test 2)	NR NR	100	Healthy pregnancies	No	NR	27–32 (test 1) 33–36 (test 2)	NR NR
Saldeen (1996) [40] Saudi Arabia case-control	UA-PI	DM type 2 GDM	21 total 2 DM type 2 9 GDM 10 impaired glucose tolerance	Pregannat women with DM type 2, GDM or impaired glucose tolerance	NR 0.079 ± 0.003% (mean ± se)	NR	NR 271.8 ± 1.9 (mean ± se)	NR NR	10	Healthy women with normal pregnancies undergoing repeated elective cesarean section	No	NR	269.0 ± 1.1 days (mean ± se)	NR NR
Grunewald (1996) [41] Sweden NR	UA-PI	DM type 1 (B, C, D, F, R)	24	Pregnant women with well-controlled insulin-dependent pregestational diabetes	Random blood glucose 5.8 mmol/L (1.8–14.3), med (range) At test I 4.2 mmol/L (1.8–8.4), med (range), At test II 5.6 mmol/L (3.6–9.4), med (range) 4.7% (3.6–7.1), med (range)	28 (19–37), med (range)	31 (29–33) (test I), med (range) 35 (33–37) (test II), med (range) 38 (35–40) (at delivery), med (range)	1st trimester65 kg (52–91) med (range) 38 gw 81 kg (69–107) med (range) NR	25	Healthy low risk pregnant women	No	27 (21–37), med (range)	31 (29–33) (test I), med (range) 35 (33–37) (test II), med (range) 39 (38–42) (at delivery), med (range)	1st trimester 59 kg (49–74), med (range) 38 gw 76 kg (60–89) med (range) NR
Weiner (1996) [42] USA NR	UA-S/D ratio	DM type 1 (B, C, D, F, R) GDM (A)	120	Well-controlled diabetics with mean blood glucose levels below 95 mg	NR NR	29.89 ± 5.4, (mean ± 2 sd) DM class A 30.2 ± 5.9 (mean ± 2 sd) DM class B–R	>30 (at assessment) DM class A 38.3 ± 1.7 (mean ± 2 sd) DM class B-R 37.7 ± 5.18(mean ± 2 sd), at delivery	NR NR	55	Non-diabetic low-risk pregnant women	No	29.4 ± 6.4	>30 (at assessment) 39.7 ± 1.4 (mean ± 2 sd)	NR NR
Ursem (1999) [43] Italy Not clear (prospective cross-sectional)	UA-PI	DM type 1 (B, C, R, F/R)	16	Well-controlled insulin-dependent diabetic pregnant women	NR 6.3% (6.1–7.1), med (range) at 1st trimestar	32 (23–32) med (range)	18 (12–21), med (range) (at assessment) 38 (30–40), med (range) (at delivery)	NR NR	16	Normal controls	Yes (gestational age)	32 (15–39), med (range)	18 (12–21) (at assessment), med (range) 40 (37–42) (at delivery), med (range)	NR NR
Boito (2003) [44] Netherlandscross-sectional	UA-PI	DM type 1 (B, C, D, R, F/R)	32	Pregestational insulin-dependent DM singleton pregnant women	NR 6.7% (4.5–12.5), mean (range)	31 (19–39) (mean-range)	25.7 (18–36), mean (range) 37.4 (28–41), mean (range)	NR NR	32	Uncomplicated pregnancies	Yes (gestational age)	31 (19–42), mean (range)	25.6 (19–36), mean (range)	NR NR
Tan (2005) [45] Malaysia case-control	UA-RI	DM type 1 DM type 2 GDM	50 Total 10 pre-existing DM 25 GDM 15 impaired glucose tolerance	Pregnant women with pre-existing DM, GDM or impaired glucose tolerance at 36 gw of amenorrhea according to the WHO 1985 criteria	NR 6.53 ± 1.14%	NR	>36 (at assessment) NR	NR NR	50	Normal pregnancies	Yes (maternal age, parity, and gestation)	NR	>36 (at assessment) NR	NR NR
Florio (2006) [46] Italy cross-sectional	UA-PI MCA-PI	GDM (A1)	13	GDM pregnancies complicated by fetal macrosomia without superimposed hypertensive disorders, preterm labor, or infection	NR NR	27.9 ± 1.1 mean ± se	40.1 ± 0.2 (at delivery)	NR NR	40	Uneventful, term gestation and delivery of a healthy infant	No	28.7 ± 1.2	39.3 ± 0.1 (at delivery)	NR NR
Girsen (2008) [47] Finland cross-sectional	UA-NR	DM type 1 (B, C, D, F)	32 Total 22 Group 1 10 Group 2	2 groups according to the HbA1c value in the 1st trimester. Group 1 (good glycemic control—HbA1c < 7.5% Group 2 (poor glycemic control–HbA1c ≥ 7.5%)	NR Group 1: 6.5 ± 0.7, 1st trimester 6.0 ± 0.8, 2nd trimester 5.9 ± 0.8 3rd trimester Group 2: 8.6 ± 0.8, 1st trimester 7.5 ± 0.8, 2nd trimester 7.3 ± 0.5 3rd trimester	Group 1 31 (18–44), med (range) Group 2 29 (21–39), med (range)	Group 1 37.3 ± 2.1 (at delivery) Group 2 36.8 ± 1.7 (at delivery)	NR NR	60	Healthy, non-diabetic women after uncomplicated pregnancy and delivery	No	NR	40.4 ± 1.2 (at delivery)	NR NR
Russell (2009) [48] Ireland prospective study	UA-PI	DM type 1 (B, C, D, F, R, F/R)	45	Pregnant women with pregestational diabetes lasting for 16.5 ± 8.7 years.	NR 7.5 ± 1.5% Early pregnancy 6.6 ± 0.9% at 14 gw 6.2 ± 0.8% at 20 gw 6.3 ± 0.8% at 36 gw	32 ± 4	38 ± 1 (at delivery)	NR 26.13 ± 4.34 kg/m^2^,	39	Uncomplicated pregnancies with no evidence of impaired glucose tolerance, without glycosuria during their pregnancy or any other indication for formal glucose tolerance testing	No	32 ± 5	39 ± 1 (at delivery)	NR 22.97 ± 3.57 kg/m^2^,
To (2009) [49] China prospective study	UA-PI	GDM	78 Total 16 GDM 62 IGT	Pregnant women before 24 gw with risk factors for GDM such as advanced age (>35 years at expected date of confinement), obesity (BMI > 25), family history of type I or type II diabetes, significant obstetric history of previous GDM, previous fetal macrosomia, or previous unexplained stillbirths.	NR NR	33.1 ± 5.4	38.3 ± 1.15 (at delivery)	NR NR	62	Non-diabetic non-hypertensive patients between 36 and 40 gw randomly selected during the same study period when they were scanned for placental location, fetal size, or liquor volume or fetal presentation	No	30.8 ± 5.0	38.9 ± 1.41 (at delivery)	NR NR
Parlakgumus (2010) [50] Turkey prospective study	UA-S/D ratio	DM type 1 DM type 2 GDM	20	Pregnant women with pre-gestational and gestational DM	NR NR	33.2 ± 4.18	37.2 ± 2.25 (at delivery)	71.6 ± 7.4 kg NR	25	Healthy pregnant women whose 50 g glucose tolerance test at 24 weeks was found to be normal	No	34 ± 4.24	38.6 ± 1.52 (at delivery)	69.4 ± 6.9 kg NR
Turan (2011) [51] USA prospective study	UA-PI	DM	63	Insulin-dependent pregestational DM with moderate to poor glycemic control	NR 7.5% (5.1–12.7), med (range)	32.5 ± 6.68	12.5 ± 0.59 (at assessment)	NR 32.6 kg/m^2^ (19–61), med (range)	63	Pregnant women without DM	Yes (gestational age, UA and DV indices)	32.1 ± 6.03	12.6 ± 0.55 (at assessment)	NR 25.0 kg/m^2^ (17–42), med (range)
Nanda (2012) [52] UK prospective study	UtA-PI	GDM	60	Pregnant women between 11^+0^ and 13^+6^ gw with GDM attending routine first pregnancy control visit	NR NR	32.0 (28.5–35.6), med (IQR)	89.1 days (86.2–93.1) (at assessment), med (IQR) 38.5 (38.1–39.6) (at delivery), med (IQR)	76.5 kg (64.3–94.0), kg, med (IQR) 28.6 kg/m^2^ (24.6–34.2), med (IQR)	240	Pregnancies with no medical complications, such as hypertensive disorders or diabetes mellitus, resulting in the birth after 37 weeks’ gestation of phenotypically normal neonates with birth weight between the 5th and 95th percentiles for gestational age	Yes (NR)	33.0 (27.3–35.9), med (IQR)	88.9 days (86.1–91.2) (at assessment), med (IQR) 39.7 (38.6–40.5) (at delivery), med (IQR)	64.0 kg (58.9–70.0), med (IQR) 23.8 kg/m^2^ (21.7–26.2), med (IQR)
Fouda (2013) [53] Egypt prospective study	UA-RI	DM type 1 DM type 2	69 Total 23 Pre-gestational DM 22 GDM 24 DM + HTA	Pregnant women with high (maternal age above 35 years, obesity, family history of diabetes mellitus, glycosuria, past history of gestational diabetes, infant macrosomia and unexplained stillbirth) and low risk after the first antenatal visit.	NR 5.66 ± 0.8%	26.35 ± 2.6	37.21 ± 0.75 (at delivery)	NR NR	27	Uncomplicated pregnancies	No	25.96 ± 2.18	37.69 ± 0.75 (at delivery)	NR NR
Suranyi (2013) [54] Hungary case-control study	UA-PI	DM type 1 (B, C, D)GDM (A1, A2)	99 Total 43 DM 56 GDM	DM type I with good glycemic control (HgA1c: 20–42 mmol/mol)	NR NR	DM 32 ± 5 GDM 33 ± 5.1	DM 31 ± 7^+4^ (at assessment) GDM 30^+6^ ± 6^+4^ (at assessment)	NR NR	113	Non-pathological control group	No	30.7 ± 5.4	28^+4^ ± 5^+5^ (at assessment)	NR NR
Savvidou (2013) [55] UK prospective study	UtA-PI	GDM	1037	Pregnant women attending their routine first hospital visit between 11^+0^ and 13^+6^ gw	NR NR	32.8 ± 5.4	89.2 ± 4.2 days (at assessment) 38.6 ± 1.4 (at delivery)	NR 29.9 ± 6.7 kg/m^2^	56 649	Normoglycemic controls	No	30.7 ± 6.0	89.0 ± 4.1 days (at assessment)	NR 25.4 ± 5.1 kg/m^2^
Shabani Zanjani (2013) [21] Iran cross-sectional study	UA-PI UA-RI UA-S/D ratio MCA-PI right and left MCA-RI right and left MCA-S/D ratio right	GDM	33	Singleton pregnant woman with at least 24 gw without any history of DM, PE, renal diseases, blood disorders, and hyperlipidemia	113.50 ± 25.03 mg/dL NR	31.21 ± 5.94	34.46 ± 2.62 (at assessment) NR	NR NR	33	The non-GDM pregnant women selected from the same perinatology clinic during the same period of time	Yes (gestational age)	26.31 ± 7.59	34.64 ± 3.24	NR NR
Li (2014) [56] China prospective cohort study	UA-PI	GDM	226	Pregnant GDM Chinese women who delivered babies at the obstetric department of the first affiliated hospital	NR NR	29.48 ± 3.54	274.70 ± 8.03 days (at delivery)	52.57 ± 7.13 kg, prepregnancy 68.16 ± 8.58 kg (at delivery) 20.64 ± 2.46 kg/m^2^, prepregnancy	519	Non-GDM pregnant women	No	28.32 ± 3.52	274.42 ± 9.69 days (at delivery)	51.58 ± 6.79 kg (prepregnancy) 67.76 ± 7.93 kg (at delivery) 20.11 ± 2.33 (prepregnancy)
Gonzales Gonzales (2014) [57] Spain Not clear (prospective case-control study)	UtA-PI	DM type 1 DM type 2	69 Total 44 DM type 1 25 DM type 2	Pregnant women with pregestational DM undergoing 1st trimester combined screening for aneuploidies	NR 6.50 ± 0.87%	32.5 ± 4.6	11–13 (at assessment) 273 days (266–280), med (IQR) (at delivery)	78.4 ± 17.0 kg 29.2 ± 5.7 kg/m^2^	94	Cases without pregestational diabetes	Yes (maternal characteristics in terms of chronic hypertension, obesity and smoking status)	30.7 ± 6.4	281 days (274, 286), med (IQR) (at delivery)	73.5 ± 15.0 kg 27.9 ± 5.4 kg/m^2^
Moran (2014) [58] Ireland prospective cohort study	UA-PI MCA-PI UtA-PI	DM type 1 DM type 2	50 Total 37 DM type 1 13 DM type 2	Pregnant women with pregestational type 1 and type 2 DM	NR NR	33 (21–45) *n* (range/%)	12^+2^ to 39^+5^ (at assessment)	NR 24.43 kg/m^2^ (18.44–79.8), mean (range)	250	Normal controls defined as no pv bleeding at any stage in the pregnancy, no medical disorder requiring treatment, e.g., diabetes, or any degree of hypertension, fetal anomaly or a suspicion or diagnosis of intrauterine growth restriction	No	31 (16–44), *n* (range/%)	12^+6^ to 39^+5^ (at assessment)	NR 25.43 kg/m^2^ (16.16–50.97), med (range)
Bhorat (2014) [59] South Africa Not clear (prospective cross-sectional study)	UA-RI MCA-RI	GDM (A2)	29	Women with suboptimally to poorly controlled insulin-dependent GDM diabetes in the 3rd trimester	11.9 mmol/L (8.3–15.9), med (IQR) NR	32 (30–33), med (IQR)	35 (34–36), med (IQR) (at assessment) 38.35 (37.71–38.71), med (IQR) (at delivery)	NR NR	29	Normal pregnancies	Yes (gestational age, maternal age)	32 (30–33), med (IQR)	35 (34–36), med (IQR) (at assessment) 39.43 (39–39.71), med (IQR) (at delivery)	NR NR
Pala (2015) [60] Turkey case-control study	UA-PI MCA-PI	GDM	39	Singleton pregnancies between 24 and 39 gw	NR NR	30.05 ± 5.56	34.92 ± 3.16	NR NR	42	Healthy singleton pregnancies between 24 and 39 gw	Yes (gestational age, maternal age, and parity)	29.32 ± 5.79	33.65 ± 3.64	NR NR
Liu (2016) [61] China Not clear (observational study)	UA-PI UA-RI UA-S/D ratio MCA-PI MCA-RI MCA-S/D ratio UtA-PI UtA-RI UtA-S/D ratio	GDM	147	Singleton pregnant women aged 25–38 years, between 37 and 40 gw (within 1 week before delivery) with an OGTT performed in the 2nd trimester, and gestational age calculated from the first day of the last normal menstrual period and confirmed by the 1st trimester ultrasound scans	NR NR	30.80 ± 3.00	38.0 ± 0.68 (at assessment)	73.50 ± 12.06 kg, (at assessment) 23.87 ± 3.58 kg/m^2^ (prepregnancy)	124	Normal pregnancies	No	29.94 ± 3.60	38.0 ± 0.65 (at assessment)	70.35 ± 9.35 kg, (at assessment) 22.24 ± 3.20 kg/m^2^ (before pregnancy)
Peixoto (2016) [14] Brazil retrospective cohort study	UA-PI MCA-PI UtA-PI	GDM	56	Pregnant women who underwent 3rd-trimester ultrasound exams between 26w0d and 37w6d of gestation	NR NR	27.60 ± 6.50	32.3 ± 3.1 (at assessment) 38.2 ± 1.5 (at delivery)	82.90 ± 15.50 kg 33.30 ± 7.30 kg/m^2^	684	NR	No	25.40 ± 6.30	32.7 ± 2.9 (at assessment) 37.8 ± 2.8 (at delivery)	71.90 ± 17.00 kg 27.30 ± 6.10, kg/m^2^
Farshchian (2017) [62] Iran case-control study	UtA-PI	DM GDM	40 Total 20 DM 20 GDM	Pregnant women with gestational age of 20 to 40 gw with DM or GDM. DM pregnant women had the condition for less than 5 years, without vascular diseases, and their blood glucose was under control.	NR NR	DM 37.85 ± 4.99 GDM 35.55 ± 3.63	DM 31.70 ± 3.64 GDM 31.9 ± 4.41	NR NR	20	Normal healthy mothers without hyperglycemia with gestational age between 20 and 40 gw	Yes (gestational age, maternal age)	35.55 ± 6.01	32.45 ± 3.34	NR NR
Bugatto (2017) [63] Spain prospective cohort study	UtA-PI	GDM (A1, A2)	25	Pregnant women diagnosed with GDM in the 2nd or 3rd trimester of gestation.	80.5 ± 9.4 mg/dL NR	31.4 ± 6.0	36.1 ± 0.4	NR 26.6 ± 6.0 m/kg^2^ (pregravid)	25	Non-GDM pregnant women	No	30.5 ± 4.5	36.0 ± 0.5	NR 29.06 ± 5.0 (pregravid)
Sweeting (2017) [64] Australia case-control study	UtA-PI	GDM	248 Total 89 Early GDM 138 Standard GDM	Pregnant women who had a diagnosis of GDM made at any timepoint during pregnancy, retrospectively identified by review of pathology and electronic medical records who referred for evaluation of 1st-trimester aneuploidy and PE screening at 11–13^+6^ gw	NR NR	33 (30–36), med (IQR)	All GDM women 275 days (271–280) med (IQR) Early GDM 274 days (269–280) med (IQR) Standard GDM 276 days (271–280) med (IQR) (at delivery)	All GDM 64.4 kg (58.2–75.4), med (IQR) Early GDM 64.5 kg (58.0–76.3), med (IQR) Standard GDM 64.6 kg (59.6–75.2), med (IQR) 24.5 (22.5–28.3)kg/m^2^, med (IQR) (at assessment)	732	Women with a normal OGTT or GCT at 24 to 28 gw, randomly selected based on gestational age (via measurement of first trimester fetal crown rump length on ultrasound)	Yes (NR)	32 (29–35) med (IQR)	279 days (173–285), med (IQR) (at delivery)	63.7 kg (57.4–71.7), med (IQR) (at assessment) 23.3 (21.6–26.1) kg/m^2^, med (IQR) (at assessment)
Meiramova (2018) [65] Kazahstan NR	UA-PI	GDM	61 Total 24 Mild GDM 37 Moderate GDM	Pregnant women with mild and moderate GDM severity between 18–42 gw	NR NR	32.8 ± 6.314	30–32 (I test) and first day of delivery (II test) 37.16 ± 3.348 (at delivery)	NR 31.1 ± 7.433 kg/m^2^ (pre-gravid)	39	Pregnant women with normal glucose tolerance	No	30 ± 5.432	38.85 ± 1.247 (at delivery)	NR 24.9 ± 5.434 kg/m^2^ (pre-gravid)
Moodley (2018) [66] Canada prospective study	MCA-RI UA-RI	DM GDM	43 Total 22 DM 21 GDM	Pregnant women referred to the Heart Center by their obstetricians for fetal echocardiography due to risk factors or concerns for fetal congenital heart disease, in keeping with indications established in recent guidelines for diagnosis and treatment of fetal cardiac disease	NR NR	33.3 ± 3.7	22.3 ± 2.2 (at assessment)	85.4 ± 26.3 kg, (pre-pregnancy) 32.8 ± 9.9 kg/m^2^ (pre-pregnancy)	23	Healthy pregnant women referred for a family history of congenital heart disease, teratogen exposure, difficulty viewing all structures of the fetal heart, suspicion of abnormal fetal cardiac structures on screening ultrasound, increased nuchal thickness and a finding of an echogenic foci, all with normal fetal echocardiograms on assessment	No	31.6 ± 8.2	22.2 ± 2.4 (at assessment)	60.4 ± 7.6 kg (pre-pregnancy) 23.5 ± 2.6 kg pre-pregnancy
Wong (2018) [67] Taiwan Not clear (prospective case-control study)	UtA-PI	GDM	31	Singleton pregnancies with GDM	NR	33.58 ± 4.32	12.52 ± 0.51 and 21.90 ± 0.65 (at assessment) 37.97 ± 1.89 (at delivery)	NR 25.13 ± 5.95 kg/m^2^	124	Those who passed the GCT or OGTT	No	31.72 ± 3.31	12.49 ± 0.55 and 22.01 ± 0.52 (at assessment) 38.84 ± 1.23 (at delivery)	NR 21.35 ± 3.23 kg/m^2^
Ciobanu (2019) [22] UK prospective study	MCA-PI	DM type 1 DM type 2	40 15 DM type 1 25 Dm type 2	Singleton pregnancies with DM	NR NR	NR	NR	NR NR	71,565	Pregnant women without DM	No	NR	NR	NR NR
Dantas (2019) [68] Brazil cross-sectional study	UA-PI MCA-PI	GDM	115	Singleton pregnant women presenting for prenatal follow-up who were diagnosed with GDM in 2nd or 3rd trimester referred to the outpatient pregnancy risk reference center	Fasting blood glucose 4.91 ± 0.78, mmol/L Postprandial blood glucose 6.45 ± 1.46, mmol/L 5.69 ± 0.95%	32.2 ± 6.5	2nd or 3rd trimester (at assessment) 30.1 ± 3.7 (at delivery)	NR 30.9 ± 5.4, kg/m^2^ Category-18.5–24.9-17 (14.8%) -25.0–29.9-34 (29.6%) -≥30.0–64 (55.7%)	123	Women without GDM (i.e., negative OGTT results) who were in the second or third trimester of pregnancy and attending basic healthcare units	No	30.7 ± 6.3	2nd or 3rd trimester (at assessment) 31.2 ± 2.3 (at delivery)	NR 27.0 ± 3.9, kg/m^2^ Category: -18.5–24.9 31 (25.2%) -25.0–29.9-49 (39.8%) -≥30.0–43 (35.0%)
Bhorat (2019) [69] South Africa Not clear (prospective cross-sectional study)	UA-RI	GDM	54	Women with GDM in the 3rd trimester	NR NR	NR	3rd trimester (at assessment)	NR NR	54	Randomly selected from the antenatal clinic and who were not diabetic as defined by the WHO criteria of a 2 h level < 7.8 mmol after a 75 g OGTT	Yes (gestational age, maternal age, parity, gravidity, BMI, and past obstetric history)	NR	34.05 ± 1.03 (at delivery)	NR NR
Gasiorowska (2020) [70] Poland NR	UtA-PI	DM	38	Singleton pregnancies at about 20 gw	NR 5.6 ± 0.95% (at 20 gw, at assessment)	29.8 ± 4.7	at about 20 (at assessment)	65.3 ± 14.6 kg (pregestational) 23.7 ± 5.1, kg/m^2^ (pregestational)	961	Healthy pregnant women	No	28.5 ± 5.3	at about 20 (at assessment)	66.2 ± 12.4 kg (pregestational) 24.3 ± 4.7 kg/m^2^(pregestational)
McLaren (2020) [71] USA Not clear (prospective cross-sectional study)	MCA-PI	DM GDM	30 Total 20 DM 10 GDM	Pregnant diabetic women 18–45 years old with a gestational age of 18–36 weeks	NR NR	NR	DM 218.47 ± 34.80 days (at assessment) Pregestational DM 218.15 ± 36.71 days (at assessment) GDM 219.10 ± 32.50 days (at assessment)	NR NR	34	Low risk pregnancies without DM	No	28 ± 6.1	28.8 ± 6.4 (at assessment)	NR 26.5 ± 4.0 kg/m^2^ (at assessment)
Bachani (2020) [72] India Not clear (observational study)	UA-PI MCA-PI	GDM	31	Women with GDM on treatment	NR NR	28.74 ± 4.12	35 (at assessment)	NR 26.07 ± 3.32 kg/m^2^ (at assessment)	40	Singleton uncomplicated pregnancies	No	27.22 ± 3.56	35 (at assessment)	NR 24.44 ± 2.97 kg/m^2^
Tenenbaum-Gavish (2020) [73] Israel prospective study	UA-PI	GDM	20	Women carrying a singleton viable gestation when undergoing combined first trimester screening for aneuploidy with GDM managed either by diet (GDMA1) or treated by glyburide or insulin (GDMA2)	NR NR	33.4 (30.7–36.1) mean (95%CI)	at 11^+0^ to 13^+6^-12.7 (12.3–13.1) mean (95%CI) (at assessment) 39.0 (38.3–39.6) mean (95%CI)	NR 30.0 kg/m^2^ (27.0–33.0) mean (95%CI), (at assessment)	185	Normal pregnancies delivering a healthy baby at term	No	31.0 (30.3–31.6) mean (95%CI)	12.6 (12.5–12.7) mean (95%CI) (at assessment) 39.6 (39.4–39.8) mean (95%CI) (at delivery)	NR 23.3 kg/m^2^ (22.8–23.9) mean (95%CI) (at assessment)
Lehtoranta (2020) [74] Finland Not clear (prospective case-control study)	UA-PI MCA-PI	DM type 1	33	Pregnant women recruited consecutively at the University Hospital outpatient maternity clinics during their first visit	NR NR	28.5 ± 4.9	Between 34^+2^ and 40^+2^ (at assessment) 37.4 ± 1.5 (at delivery)	NR 26.1 ± 4.9, kg/m^2^ (prepregnancy)	67	Healthy singleton pregnancies from outpatient maternity clinics with BMI < 30 kg/m^2^, major serious illnesses and with normal 2 h oral glucose tolerance test at 24–28 gw	No	28.0 ± 4.0	Between 34^+2^ and 40^+2^ (at assessment) 39.5 ± 1.9 (at delivery)	NR 23.2 ± 3.4 kg/m^2^ (prepregnancy)
Phadungkiatwattana (2021) [75] Thailand Not clear (prospective observational cross-sectional study)	UA-PI MCA-PI	DM GDM	138 Total 46 DM 92 GDM	Pregnant women with DM (pregestational with insulin usage and gestational with diet control)	NR 5.8% (5.3–6.3) med (IQR)	33.8 ± 5.4	Between 35 and 37, 36.05 ± 0.8 (at assessment) 38.33 ± 1.08 (at delivery)	NR 25.35 ± 5.10 kg/m^2^ (pregestational)	149	Healthy pregnant women	No	29.0 ± 6.0	Between 35 and 37, 36.05 ± 0.8 (at assessment) 38.78 ± 1.1 (at delivery)	NR 22.64 ± 3.72 kg/m^2^ (pregestational)
Wei (2021) [18] China NR	MCA-PI MCA-RI MCA-S/D ratio UA-PI UA-RI UA-S/D ratio UtA-PI UtA-RI UtA-S/D ratio	GDM	76	Pregnant women with GDM admitted to the obstetric outpatient clinic	NR NR	28.71 ± 4.62	27.88 ± 2.31 (at assessment)	NR NR	76	Healthy pregnant women	No	28.62 ± 4.55	26.37 ± 2.35 (at assessment)	NR NR
Zhang (2021) [76] China NR	MCA-PI MCA-RI MCA-S/D ratio	GDM	80	Pregnant women diagnosed as having GDM treated in the hospital	NR NR	NR	at 25th–28th (at assessment) 34.17 ± 3.88	NR NR	80	Healthy pregnant women	No	NR	38.66 ± 2.75 (mean ± sd), at delivery	NR NR
Alanyali (2021) [77] Turkey prospectivecontrolled clinical trial	UA-PI	DM type 1 DM type 2	30	Outpatient pregnant women aged 18–45 years, between 24 and 26 gw according to the last menstrual period diagnosed pregestational DM type 1 or type 2	NR NR	32.00 ± 4.99	24.57 ± 0.62 (at assessment)	NR NR	30	Singleton healthy non-PE pregnant women aged 18–45 years of age without pregestational DM or additive diseases (HTA, cardiac disease, thyroid disorders, systemic lupus erythematosus) with fetus without congenital malformations	No	27.53 ± 5.22	24.53 ± 0.77 (at assessment)	NR NR
Mecacci (2021) [78] Italy case-control study	UA-PI	DM type 1	244	Pregnant women with DM recruited before 10^th^ gw	NR NR	28.3 (22–41) med (range)	16th, 20th, and 24th (at assessment)	NR NR	488	Singleton pregnant women with normal glucose tolerance test, and delivery after 20 gw followed up in the same maternal-fetal outpatient unit	Yes (race, maternal age, pre-pregnancy BMI, nulliparity, weight gain during pregnancy in ratio 1:2)	29.4 (16–41) med (range)	16th, 20th, and 24th (at assessment)	NR 23.7 (19.4–27.8) kg/m^2^ med (range)
Liu (2021) [79] China NR	MCA-PI MCA-RI MCA-S/D ratio	GDM	1268	GDM pregnant women	NR NR	31 mean	38 mean (at assessment and delivery)	NR NR	10,922	Non-GDM pregnancies	No	30 mean	39 mean (at delivery, at assessment)	NR NR
Fatihoglu (2021) [80] Turkey prospective study	UA-PI UA-RI UA-S/D ratio MCA-PI MCA-RI MCA-S/D ratio	GDM	60	GDM pregnant women	NR NR	32 (20–46) med (range)	at 18–22 (at assessment)	NR 30 (24–35) kg/m + med (range)	61	Healthy controls	Yes (gestational age)	26 (18–38) med (range)	at 18–22 (at assessment)	NR 28 (24–32) med (range)
Chen (2021) [81] China cross-sectional study	UA-PI	GDM	30	Singleton diabetic pregnancies at 24–40 gw	Fasting blood glucose: 4.78 (3.64–7.41) mmol/L, med (IQR) 1 h plasma glucose: 10.45 (3.32–14.62) mmol/L, med (IQR) 2 h plasma glucose: 9.10 ± 1.73 5.50 (4.80–6.70), med (IQR)	31.00 ± 2.92	36–40 gestational weeks (at assessment) 39.30 (37.20–40.1) med (IQR) 39.30 (37.20–40.1) med (IQR) (at delivery)	NR 21.76 (17.80–27.58) kg/m^2^, med (IQR)	31	Healthy pregnant mothers	No	29.84 ± 3.07	Fasting blood glucose: 4.29 (3.88–4.94) mmol/L med (IQR) 1 h plasma glucose: 6.91 (3.92–9.80) mmol/L med (IQR) 2 h plasma glucose: 6.46 ± 1.18 mmol/L	NR 21.00 (17.97–29.69) kg/m^2^ med (IQR)
Ali Hassan (2021) [82] Egypt prospective study	UA-PI UA-RI MCA-PI MCA-RI	GDM	25 GDM 25 GDM + PE	Singleton pregnant women in 3rd trimester (28–39 gw) with age between 25–38 years with GDM, and combined GDM with PE defined as SBP > 140 mmHg and DBP > 90 mmHg)	NR NR	25–38 (range)	28–39 (at assessment)	NR NR	25	3rd-trimester pregnant women of a single fetus between 28 and 39 gw without factor, checked by measuring fasting plasma glucose concentration < 140 mg/dL and HbA1c < 6.5%.	No	25–38 (range)	28–39 (at assessment)	NR NR
Jamal (2021) [83] Iran prospective cohort study	UA-PI UA-RI UA-S/D ratio MCA-PI MCA-RI MCA-S/D ratio	GDM	123	Pregnant women newly diagnosed with GDM at 24–28 gw treated with insulin or managed with diet	NR NR	31.5 ± 5.4	37–40 (at assessment) 38.6 ± 0.8 (at delivery)	NR NR	123	Women without GDM	No	29.7 ± 5.6	37–40 (at assessment) 38.9 ± 0.8 (at delivery)	NR NR
Perez-Martin (2022) [84] Spain Not clear (prospective and cross-sectional case control study)	UA-PI MCA-PI UtA-PI	GDM	56	GDM pregnancies	83.6 ± 9.0 mg/dL 4.9 ± 0.3%	35.5 ± 4.1	28–32 (at assessment) 38.6 ± 1.5 (at delivery)	74.1 ± 18.9 kg (pregestational) 28.2 ± 6.2 kg/m^2^ (pregestational)	65	Physicologic pregnancies with normal glucose screening that were seen during the growth scan at 28–32 gw	No	33 ± 5	30 ± 1.5 (at assessment) 39.3 ± 1.2 (at delivery)	66.5 ± 13.2 kg (pregestational) 25.1 ± 4.6 kg/m^2^ (pregestational)
Chatzakis (2022) [85] Greece cross-sectional study	UA-PI UtA-PI	GDM	25	GDM pregnancies	NR NR	32.4 ± 4.0	32 ± 2.5 (at assessment)	NR 27.3 ± 7.9 kg/m^2^ (prepregnancy) 30 ± 5.7 kg/m^2^ (at assessment)	25	Uncomplicated pregnancies	Yes (pre-pregnancy BMI, maternal age, and gestational age)	30.4 ± 6.2	31 ± 3.2 (at assessment)	NR 25.1 ± 5.2 kg/m^2^ (prepregnancy) 28.6 ± 5.0 kg/m^2^ (at assessment)
Karaca Kutulmus (2022) [86] Turkey cross-sectional	UA-PI MCA-PI	GDM	45	GDM pregnant women with poor glycaemic control and appropriate-for-gestational-age or macrosomic fetuses between 28 and 39 gw	NR NR	30.04 ± 5.33	NR 33.13 ± 2.96 (at delivery)	NR NR	49	Healthy pregnant women on routine prenatal care with the appropriate-for-gestational-age fetuses between 29 and 41 gw	Yes (gestational age)	28 ± 4.91	33.40 ± 3.22 (at delivery)	NR NR

^a^ maternal age is reported in years as mean ± sd, if otherwise then it is indicated; ^b^ gestational age is reported in gestational weeks (gw) as mean ± sd, if otherwise then it is indicated. Abbreviations: BMI—body mass index, DM—diabetes mellitus, DV—ductus venosus, DBP—diastolic blood pressure, GCT—glucose challenge test, GDM—gestational diabetes mellitus, HTA—hypertension, HbA1c—glycosylated hemoglobin, IGT—impaired glucose tolerance, NR—not reported, OGTT—oral glucose tolerance test, PE—preeclampsia, PIH—pregnancy-induced hypertension, SBP—systolic blood pressure.

## Data Availability

All additional data are available as Appendix A.

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
