# Peer review of "Doppler Indices of the Uterine, Umbilical and Fetal Middle Cerebral Artery in Diabetic versus Non-Diabetic Pregnancy: Systematic Review and Meta-Analysis"

_medicina, 2023, doi:10.3390/medicina59081502_

Round 1

Reviewer 1 Report

Diabetes mellitus is very important disease for obstetric care. In this study, the authors aimed to assess the differences in materno-fetal Doppler indices of diabetic versus non-diabetic pregnancies. This study may be contributed to literatur. After some revisions the manuscript can be reconsidered for acceptance.

1. Materno-fetal Doppler parameters can be affected not only by diabetes but also by many factors. For example; A history of additional chronic diseases, preeclampsia and/or fetal growth retardation, drug use that may affect Doppler parameters. Although some of these factors are stated as exclusion criteria in some studies, could the absence of such a feature in some studies affect the results of meta-analysis?

2. Some patients with DM are followed by using insulin, and some are followed only by appropriate diet. How might this affect the results of the study?

3. How do the authors interpret that there is no significant difference between the DM and non-DM groups in terms of fetal MCA Doppler parameters?

4. Although the pregnancy trimesters are specified, is it possible that the differences between the gestational weeks of Doppler applied in the studies affect the results?

Author Response

Q1: Materno-fetal Doppler parameters can be affected not only by diabetes but also by many factors. For example; A history of additional chronic diseases, preeclampsia and/or fetal growth retardation, drug use that may affect Doppler parameters. Although some of these factors are stated as exclusion criteria in some studies, could the absence of such a feature in some studies affect the results of meta-analysis?

A1: Thank you for this comment. Following sentences is introduced in the discussion:

"However, materno-fetal Doppler parameters can be affected not only by DM but by many other factors. Although some of these factors like chronic diseases, preeclampsia, fetal growth retardation, and drug use, that may affect Doppler parameters, are stated as exclusion criteria in some studies, the absence of such exclusion criteria (or not reporting them) in others may affect the results of our meta-analysis. For example, Fouda et al. found that HgA1c was higher in pregestational diabetic women with chronic hypertension. Also, UA-RI was higher in diabetic pregnancies with hypertension, but not in diabetic pregnancies without hypertension, in comparison to uncomplicated pregnancies as controls [53]. Hassan et al. reported higher UA-PI levels in diabetic pregnancies complicated by preeclampsia [82]."

and in limitations:

"First, the absence of exclusion criteria such as additional chronic diseases, preeclampsia and/or fetal growth retardation, and drug use in some of the included studies may affect the overall pooled estimate of this meta-analysis."

Q2. Some patients with DM are followed by using insulin, and some are followed only by appropriate diet. How might this affect the results of the study?

A2: Thank you for this comment. Following sentences is now given in the Introduction:

In a large meta-analysis including 7,5 million of pregnancies, GDM was significantly associated with a range of adverse pregnancy outcomes [4]. Women with GDM and no insulin use, have increased odds of caesarean section, preterm delivery, macrosomia, infant born large for gestational age and low Apgar score, while for women with GDM using insulin, the odds of having an infant requiring admission to the neonatal intensive care unit, who is large for gestational age, with respiratory distress syndrome, and/or neonatal jaundice were higher than in those without GDM [4].

and in limitations:

Second, some patients with DM included in studies, are followed by using insulin, and some are followed only by appropriate diet. This broad range of therapy regimens might also affect the results of the meta-analysis.

Q3. How do the authors interpret that there is no significant difference between the DM and non-DM groups in terms of fetal MCA Doppler parameters?

A3: Thank you for this comment. Following sentences is now given in the Disscusion:

"Results of our meta-analysis presented no significant differences between the DM and non-DM groups in terms of fetal MCA Doppler parameters. It is known that long-term uncontrolled hyperglycemia, chronic hypertension, preeclampsia, IUGR can lead to placental vascular dysfunction with changes even of fetal circulation [53, 82, 104-106]. The effect of metabolic changes due to diabetes mellitus during pregnancy on the fetus may be acidemia but without hypoxemia, thus that redistribution seen in fetal hypoxemia may not occur even in severely compromised fetuses and therefore it is of huge importance not to mislead this state by apparently normal fetal Doppler results [80]."

Q4. Although the pregnancy trimesters are specified, is it possible that the differences between the gestational weeks of Doppler applied in the studies affect the results?

A4: Thank you for this comment. Following is now stated in the Limitations:

"Third, although the pregnancy trimesters are specified in some studies, it possible that the differences between the gestational weeks of Doppler measurements applied in the studies affects the overall results."

Reviewer 2 Report

Dear Authors 

I read the paper: Doppler indices of the uterine, umbilical and fetal middle cerebral artery in diabetic versus non-diabetic pregnancy: systematic review and meta-analysis, which falls whithin the aim of Medicina. Honestly, the topic is interesting enough to attract the readers' attention.

I have some specific recommendations around the study methodology and reporting that I describe under each section:

 INTRODUCTION

1) Line 47: Cesarean section is usually needed to deliver new-born, but baby is also at increased risk of health problems after the birth.

The present declaration lacks completeness and correctness. In pregnancies complicated by gestational diabetes mellitus (GDM), it is common to consider labor inductions based on glycemia status. However, I recommend that the author correct this erroneous statement and provide supporting paragraphs and citations, referencing a recent article: 

- DOI https://doi.org/10.1016/j.ajog.2015.12.021

  • DOI: https://doi.org/10.1007/s00404-019-05183-z

2) Line 58: During a physiological pregnancy, spiral remodeling modifies arteries from low-flow/high-resistance to high-flow/low-resistance vessels.

In light of the study on uterine arteries, I propose that the authors include a reference to their groundbreaking discovery concerning this subject matter: 

DOI: 10.1055/a-2075-3021 (https://www.thieme-connect.com/products/ejournals/abstract/10.1055/a-2075-3021)

3) line 54: Before to tell abut the fetal complication, the authors should write about a correct pregnant alimentation and maternal weight gain that could modify the hyperglycemia status and reduce the possible complication. I suggest the following article: 

  • doi: 10.1097/EDE.0000000000000629
  • DOI: 10.1097/01.ede.0000142151.16880.0
  • https://doi.org/10.1111/jog.15205

4) DISCUSSION 

The authors should provide a more comprehensive analysis of the strengths and limitations of their study. It is recommended to clarify and improve these aspects more effectively.

In conclusion, Researchers have composed the conclusion and discussion section very well. The meta-analysis can be read in a fluent and engaging manner. The title reflects the focus of the analysis. Paper's theoretical framework is very clear and it fits into the literature. I suggest a minor revision before the publication. 
